# Southwestern national park service employee risk, knowledge, and concern for triatomine exposure: A qualitative analysis using a novel knowledge, attitudes, and practices survey

**Antonio Alvarado** [1,2]*, **Emily M. Mader**[1,2], **Danielle Buttke**[3], **Laura C. Harrington**[1,2]

**1** Department of Entomology, Cornell University, Ithaca, New York, United States of America, **2** Northeast Regional Center for Excellence in Vector-borne Diseases, Ithaca, New York, United States of America, **3** Office of Public Health and Wildlife Health Branch, National Park Service, Fort Collins, Colorado, United States of America

\* aa2757@cornell.edu

**Data Availability Statement:** All relevant data are within the manuscript and its Supporting Information files.

## Abstract

Chagas disease (CD), caused by the parasite *Trypanosoma cruzi*, is a neglected parasitic infection in the United States (US). In the Southwestern US, National Park Service (NPS) employees are a unique population with potential exposure to CD. This population lives in close contact with several species of sylvatic triatomine bugs, the vectors of *T. cruzi*, that may enter residential buildings at night. Despite the higher potential risk of CD transmission for southwestern NPS employees, the socio-cultural factors that impact autochthonous CD transmission in the US remain unknown. To address this gap, we investigated how NPS employee knowledge and attitudes impact their triatomine preventive behaviors. We distributed a 42-item online questionnaire to NPS employees at four national parks in Arizona and Texas. We detected high self-reported bite exposure in NPS housing, despite moderate- to high-frequency of prevention behaviors. Specific behaviors, such as often or always repairing window screens, were associated with a decreased risk of putative triatomine bug exposure. Additionally, NPS employees had low knowledge of CD. For those with greater knowledge of CD, it was not associated with increased frequency of prevention behavior. We found that increased CD anxiety was associated with increased personal agency to reduce the risk of CD. These results demonstrate the influence of knowledge and attitudes regarding CD on triatomine prevention behavior within a potential high-risk population in the US, and the importance of utilizing strategies beyond provision of education to influence behaviors.

## Author summary

Chagas disease (CD), a neglected vector-borne disease, negatively impacts 300,000 United States citizens in present time. The parasite that causes CD, *Trypanosoma cruzi*, is spread through the infected feces of triatomine bugs. Vector-borne transmission risk is

**Funding:** This work was supported by the Centers for Disease Control and Prevention (CDC) through Cooperative Agreement Number 1U01CK000509 between the CDC and the Northeast Regional Center for Excellence in Vector Borne Diseases awarded to LCH. The funders had no role in study design, data collection and analysis, decision to publish, or preparation of the manuscript.

**Competing interests:** The authors have declared that no competing interests exist.

considered highest in the southwestern United States, where there is greater triatomine diversity. Southwestern National Park Service (NPS) employees are a unique population regarding human-CD risk because they live in close contact with several species of triatomines. However, CD transmission does not solely result from biological factors; sociocultural factors, including what human populations understand and what they do in response to a disease, are integral for vector-borne agent spread. Therefore, we investigated how NPS employee knowledge and attitudes impact their triatomine preventive behaviors. We detected high self-reported triatomine bug bite exposure in southwestern national parks. Additionally, NPS employees overall had low knowledge of CD, but greater knowledge of CD was not associated with better practices. Human emotions, behavior, and environmental factors are deeply rooted within the CD transmission cycle. This research adds to the growing body of literature on the CD knowledge, attitudes, and practices of a high-risk population in the US.

## Introduction

Chagas disease (CD) is caused by the kinetoplast, protozoan parasite *Trypanosoma cruzi*, discovered by Carlos Chagas in 1909 [1]. There are several routes of parasite transmission including via triatomine bugs vectors through their infected feces, blood-borne, oral, congenital, and organ-derived [2]. Following the transmission of *T. cruzi* to a human host, CD is characterized into two clinical phases: acute and chronic [3,4]. The acute infection phase can last 2 to 3 months from initial infection, which may manifest as flu-like symptoms, febrile illness, enlarged lymph nodes, and skin rashes [2,5,6]. Since these are common symptoms shared by many diseases, individuals may not seek medical care nor receive diagnostic testing. Thus, an estimated 20% to 30% of infected persons progress into the chronic form of CD, months to years after the initial infection [2,6]. Despite the lack of symptoms, chronic CD is dangerous because once symptoms do manifest, *T. cruzi* becomes more difficult to eliminate and the parasite can cause severe cardiomyopathy and gastrointestinal complications [6].

Despite the severity of the disease, there are currently an estimated 300,000 persons living with CD in the United States (US). Additionally, there are currently approximately 30,000 to 45,000 CD-cardiomyopathy cases in California, Texas, Florida, New York, and Arizona [7,8]. Although most CD cases are imported, and autochthonous CD in the US is rare, vector-borne transmission may occur at higher rates than previously suspected in the southwestern region. For instance, since CD became reportable in Texas in 2013, there have been 34 reported autochthonous CD cases [9]. In addition to Texas, CD is only reportable in five other states including Arizona, Arkansas, Louisiana, Mississippi, and Tennessee [10]. CD is now becoming an important public health issue as humans expand into habitats of the triatomine vector and animal reservoirs, putting individuals at increased risk for *T. cruzi* infection.

Triatomines, also known as kissing bugs or conenose bugs, are nocturnal, obligate hematophagous insects found almost exclusively in North and South America [2]. There are 11 species of triatomines widely distributed across the US. However, the greatest species diversity occurs in the southwestern US. Most efficient triatomine species defecate shortly after their bloodmeal; the shorter the timespan between feeding and the first defecation, the higher the risk of infection [11,12]. In the southwest, there are also high *T. cruzi* infection rates reported in triatomines [13]. This level of triatomine diversity and infection poses a potential risk for hundreds of seasonal and permanent residents who visit, live or work on National Park Service (NPS) lands within the southwest.

National parks provide a unique environment where wild animals, vectors, and humans have the potential to come into contact. Housing structures within these parks can be in close association with pack rat (*Neotoma* spp.) nests where triatomines may live in the nesting material, and at times, these nest-associated triatomines can enter homes and bite humans. Pack rats are the primary wildlife reservoirs of *T. cruzi* in the southwestern US [14]. Triatomines may obtain bloodmeals from infectious pack rats and transmit the parasite to humans. Additionally, although significant recent investments have been made in restoring NPS housing and infrastructure, it is reported that some NPS housing, which was largely built during the 1930s, 1960s, and 1980s, is deteriorating and of poor-quality, which may make residences more susceptible to triatomine intrusion [15–17]. While it is hypothesized that triatomines in the southwestern deserts are not domesticated, they are attracted to lights in homes during their dispersal flights, which increases their exposure to humans. This behavior may lead them to regularly bite residents in the southwest [18–22]. For these reasons, NPS employees may be at a higher risk for CD than other populations in the US.

Despite the potential high risk for triatomine exposure among NPS employees that reside in the southwestern national parks, there are no validated surveys to assess the knowledge, attitudes, and practices (KAP) and the risk of NPS employees regarding CD. In fact, few studies exist that assess CD KAP of US residents. This information is beneficial to identify solutions for improving the knowledge and prevention of CD; potential knowledge gaps in an individual's understanding and response to vector exposure risk; as well as barriers for vector control efforts [23–27]. Identifying the gaps in knowledge and practices among NPS employees, as well as potential control measures, will enable the NPS to implement more effective CD behavior change campaigns and mitigation efforts. We sought to address the gap in understanding CD knowledge and preventive practices among NPS employees by determining 1) the influence of NPS employee knowledge and attitudes on their triatomine prevention behaviors, 2) the triatomine prevention and control measures that NPS employees are currently using and willing to use, 3) self-reported human and triatomine exposure frequency, and (4) the risks to NPS employees.

## Methods

### Ethics statement

Protocols and procedures involving human subjects were reviewed by the Institutional Review Board of Cornell University, Protocol #: 2102010140, and deemed exempt.

### Study design and sampling framework

**Questionnaire development.** To determine self-reported exposure to triatomines and to assess KAP regarding CD of NPS employees, we implemented an anonymous questionnaire using Qualtrics. The KAP survey was distributed in both English and Spanish. It comprised 42 questions, which involved a combination of closed-ended questions with ordinal response scales (five-point Likert-type response scales), closed-ended questions with categorical response options, and closed-ended questions with multiple response options. It is reported that multiple response items and ordered response scales are effective methods to objectively assess knowledge [28]. The questionnaire was divided into five sections: demographics (8 questions), knowledge (12 questions), attitudes (6 questions), practices (5 questions), and personal experiences with triatomines and small mammals in NPS housing (11 questions) (S1 File).

Knowledge questions included familiarity with acute and chronic CD signs and symptoms, CD treatment, CD transmission, triatomine identification, and the animal reservoir hosts.

Participants could not go back and change their answers to questions. After answering the knowledge questions, a paragraph of educational information regarding CD in the US was presented to participants. Attitude questions included perceived severity of CD, perceived risk for developing CD while working with the NPS, perceived confidence in identifying triatomines, and perceived effectiveness of personal and NPS triatomine control. Practice questions included the frequency of personal preventive measures against triatomines and small mammals, the sources where individuals received information on triatomines, personal behavior following a triatomine bite, and personal control measures that individuals would use against triatomines. Lastly, we asked questions related to personal experience with triatomines and small mammals in NPS housing, including seeing bugs in their homes, unexplained insect bites while sleeping, the location of bites, the frequency of bites (number of bites per year), if they owned a pet, if they found triatomines on their pet or pet's sleeping area, perceived quality of housing conditions, and if they had seen small animals or rodents within their house.

The survey was thoroughly reviewed by members from the NPS, the Parasitic Diseases Branch at the Centers for Disease Control and Prevention (CDC), and the Northeast Regional Center for Excellence in Vector-Borne Diseases at Cornell University before administration. The purpose of this content validation was to ensure the questions were not ambiguous and content was appropriate and accurate. Modifications were carried out regarding content and structure of questions based on these expert reviews. Following this content review, the survey was piloted to beta-testers within the Harrington Lab at Cornell University for usability and comprehension. The data was tabulated and reviewed for technical issues such as missing data or errors with scoring.

## Data collection

The KAP survey was administered to six park units: Big Bend National Park (BIBE), Saguaro National Park (SAGU), Organ Pipe National Monument (OPNM), and the Southeast Arizona Group (SEAZ), which includes Coronado National Memorial (CORO), Fort Bowie National Historic Site (FOBO), and Chiricahua National Monument (CHIR) (Fig 1). The target population was NPS employees that were 18 years or older and worked at these southwestern park units. These locations were selected due to the high frequency of employees reporting triatomines within their housing. BIBE, where triatomine surveillance and CD education had previously taken place, was compared with several Arizona parks where limited or no prior surveillance or educational operations were conducted. BIBE was the only park unit located within the Chihuahuan Desert ecoregion, while the five Arizona parks were located within the Sonoran Desert ecoregion.

The sampling frame was obtained by emailing NPS partners at each respective park, who then sent an estimate of their number of employees. We estimated that the survey was sent to approximately 60 NPS employees at BIBE, 50 NPS employees at SAGU, 35 NPS employees at OPNM, and 32 NPS employees from the SEAZ group. Therefore, the sampling frame was approximately 177 individuals. Determination of the sampling frame was difficult since many seasonal NPS employees leave and arrive throughout the year. To ensure a larger sample size was reached, the response rate was monitored weekly via Qualtrics and individual parks with less representation were notified to increase survey awareness among NPS employees. The targeted minimum response rate per park was 50%.

NPS administrative partners were e-mailed an internet link for the questionnaire to distribute to NPS employees who currently worked at the six national park units of interest during 20 April–27 June 2021. SEAZ and OPNM distributed the questionnaire on 20 April 2021, SAGU distributed the questionnaire on 22 April 2021, and BIBE distributed the questionnaire on 14

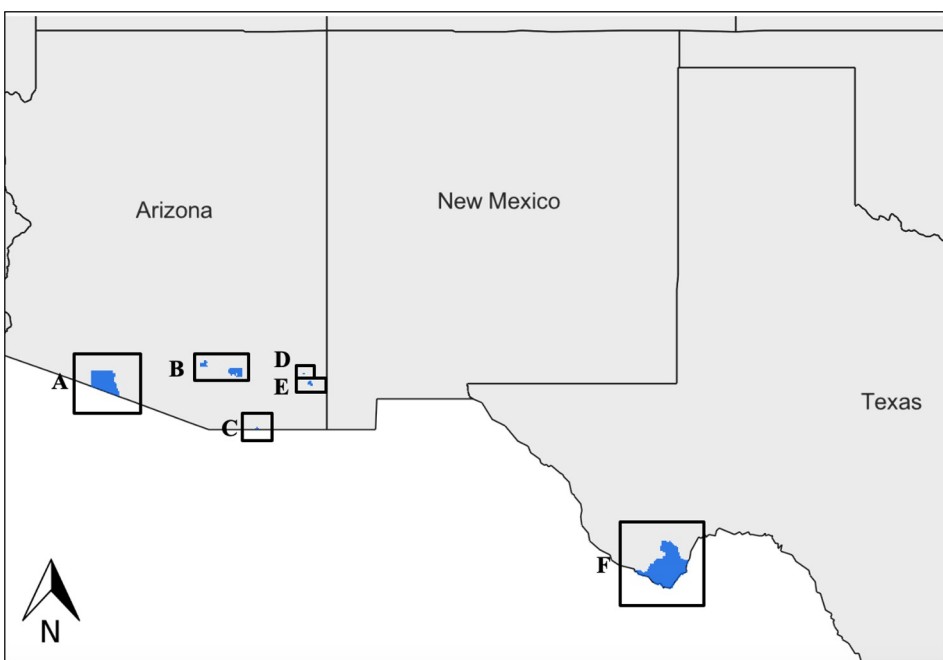

**Fig 1. Map of southwestern parks included in the survey to assess knowledge, attitudes, and practices.** (A) Organ Pipe National Monument; (B) Saguaro National Park; (C) Coronado National Memorial; (D) Fort Bowie National Historic Site; (E) Chiricahua National Monument; (F) Big Bend National Park. Figure created on ArcGIS Online with the Charted Territory basemap (https://basemaps.arcgis.com/arcgis/rest/services/World_Basemap_v2/VectorTileServer) and USA National Park Service Lands layer (https://landscape10.arcgis.com/arcgis/rest/services/USA_Federal_Lands/ImageServer).

May 2021. BIBE received the same survey as the Arizona park units, but without demographic questions due to their specific requests. The survey was open for each park for approximately 10 weeks.

## Data analysis

Analyses were conducted in RStudio 2020 (R Foundation for Statistical Computing, Vienna, Austria. URL: http://www.R-project.org/) with the following packages: tidyverse, dplyr, psych, readxl, reshape2, rstatix, ggpubr, dunn.test, MASS, and PMCMRplus [29–38].

For the knowledge section, each correct answer carried 1 mark, while wrong or "Not sure" carried 0 mark. The first knowledge question, "How familiar are you with Chagas disease signs and symptoms?", was not included in the knowledge score because it assessed the individual's perceived familiarity with CD, not knowledge of CD. We removed knowledge question 3, "What are the potential long-term health effects of chronic infection with Chagas disease?", since 44.9% (n = 40) of respondents skipped this question. Following this alteration, knowledge score was calculated out of 10 possible points.

Only the first question in the practice section was scored, which asked about the frequency of personal preventive measures using a matrix question format with nine statements. Responses of "Never" were scored "0", while "Rarely", "Sometimes", "Often", and "Always" were scored as "1", "2", "3", and "4", respectively. "Always" was assigned the highest score because it represents the maximum frequency for a preventive measure. The total score range for the practices section was 0 to 36. Additionally, for covariate analyses of the knowledge and practices scores, we restricted the responses to those who answered every question. After list-wise deletion, we retained 69 responses to explore aggregate knowledge and practices scores.

Attitude questions were ordered on a Likert scale and scored out of 5, with "1" being "Strongly disagree" and "5" being "Strongly agree." Principal components analysis (PCA) with "promax" rotation was used to reduce the dimensions of the attitudes items and generate two factor scores. The variable pertaining to whether a park provided awareness/educational materials regarding CD to their employees was generated as a binomial variable with 1 for "Yes" and 0 for "No". Internal consistency and reliability of the calculated scores were evaluated using Cronbach's alpha coefficient. Alphas 0.50 and above were considered reliable and internally consistent measures [39].

For each test, a p-value less than 0.05 was considered statistically significant. Knowledge scores by park variable were compared using ANOVA tests, while knowledge scores by demographic variables were compared using Kruskall-Wallis tests. Practice scores were compared by park and demographic variables using Kruskall-Wallis tests with post-hoc Dunn's test. Associations between knowledge, practice, and attitudes scores were assessed using Spearman's rank-order correlation. Comparisons of knowledge, practice, and attitudes scores by the awareness/educational variable were conducted with Mann-Whitney U tests or Two-Sample t-Tests.

Analyses that involved NPS housing and triatomine exposure were filtered based on respondents that correctly identified triatomines in the knowledge section. This was conducted based on the premise that if respondents cannot correctly identify a triatomine bug, they are not likely to reliably report seeing them. Following this filter, bivariable analyses with Fisher's exact tests were used to assess associations between respondents' identification abilities and various factors of interest.

## Results

### Respondent demographics

We collected 89 survey responses for an overall response rate of 50.28% from the approximated sampling frame (Table 1). Survey responses are summarized in S1 Table.

Most respondents resided and worked in NPS units in Arizona (AZ) and these parks did not offer any form of CD or triatomine awareness material (n = 60; 67.42%). Among the AZ parks, respondents were primarily male (n = 36; 60%), older than 50 years (n = 25; 41.67%) and had been employed by NPS for 10 years or longer (n = 28; 46.67%). Education ranged from high school or less (n = 4; 6.67%) to graduate school or higher (n = 20; 33.33%), but roughly half of respondents were college graduates (n = 28; 46.67%). Among all respondents, approximately half lived in NPS housing (n = 49; 55.06%), 43.82% (n = 39) worked less than

**Table 1. Survey response rates by national park and park-related information.**

| Park | Approximate National Park Service employee population size | Responses (response rate %) | Employees residing in National Park Service housing (%) | State | Park provides triatomine and/or Chagas disease awareness/education material |
|------|------|------|------|------|------|
| **Big Bend National Park** | 60 | 29 (48.33) | 28 (96.55) | Texas | Yes |
| **Saguaro National Park** | 50 | 22 (44) | 1 (4.55) | Arizona | No |
| **Organ Pipe National Monument** | 35 | 21 (60) | 10 (47.62) | Arizona | No |
| **Southeast Arizona Group** | 32 | 17 (53.13) | 10 (58.82) | Arizona | No |
| **Total** | 177 | 89 (50.28) | 49 (55.06) | | |

20 hours outdoors each week, and about half of the respondents (n = 49, 55.06%) owned pets, most of whom owned dogs (n = 31; 63.27%). Nearly all BIBE employees (n = 28; 96.55%) resided in NPS housing, while nearly all SAGU employees resided in non-NPS housing (n = 21; 95.45%). More SEAZ employees resided in NPS housing (n = 10; 58.82%) than non-NPS housing, while about half of OPNM resided in NPS housing (n = 10; 47.62%). Additionally, AZ respondents reported a wide range of satisfaction with their housing. Satisfaction ranged from not satisfied at all (n = 3; 5%) to completely satisfied (n = 9; 15%), with the largest proportion of respondents indicating neither satisfied nor dissatisfied (n = 14; 23.33%).

## NPS employees have low knowledge of CD and triatomine vectors

Overall, NPS employees had low knowledge of CD and triatomine vectors. Only 19.32% (n = 17) agreed that they were either very familiar or familiar with CD signs and symptoms, and only 27.27% (n = 24) were able to correctly identify the potential symptoms of acute CD (Table 2). Most respondents were unfamiliar with the questions relating to *T. cruzi* transmission. For example, only 7.79% (n = 6) correctly identified all modes of *T. cruzi* transmission and only 28.41% (n = 25) knew that the parasite is transmitted through triatomine feces. The pattern of knowledge question responses within the subset of respondents that answered each of the knowledge, attitudes, and practices sections (n = 69) mirrored those of the full survey sample.

Importantly, a majority of NPS employees correctly identified the triatomine bug from the other arthropods (n = 76; 86.36%), understood that triatomines are nocturnal (n = 74; 84.09%), and knew that not all triatomines are infected with the parasite (n = 59; 67.05%). Additionally, individual parks had significantly different answers for certain questions. BIBE

**Table 2. Percent of individuals choosing correct knowledge section responses by park.**

| Choices | Big Bend National Park | Southeast Arizona Group | Saguaro National Park | Organ Pipe National Monument | Overall Sample | Subset Full Responses |
|---|---|---|---|---|---|---|
| **Correctly identified triatomine** | 28/29 (96.55%) | 14/17 (82.35%) | 16/21 (76.19%) | 18/21 (85.71%) | 76/88 (86.36%) | 63/69 (91.30%) |
| **Humans are most at risk for being bitten by a triatomine at night**[*] | 29/29 (100%) | 13/17 (76.47%) | 20/21 (95.24%) | 12/21 (57.14%) | 74/88 (84.09%) | 62/69 (89.86%) |
| **All triatomines are NOT infected with** *Trypanosoma cruzi* | 23/29 (79.31%) | 11/17 (64.71%) | 13/21 (61.90%) | 12/21 (57.14%) | 59/88 (67.05%) | 48/69 (69.57%) |
| **Heart and digestive issues are potential long-term health effects of chronic CD** | 15/23 (65.22%) | 6/7 (85.71%) | 5/11 (45.45%) | 4/8 (50%) | 30/49 (61.22%) | Did not include this question in subset |
| **The presence of animal reservoirs, such as pack rats, are important factors for human risk of CD** | 14/29 (48.28%) | 10/17 (58.82%) | 14/21 (66.67%) | 6/21 (28.57%) | 44/88 (50%) | 38/69 (55.07%) |
| **CD is caused by a parasite** | 16/29 (55.17%) | 7/17 (41.18%) | 6/21 (28.57%) | 11/21 (52.38%) | 40/88 (45.45%) | 35/69 (50.72%) |
| **Dogs can get sick from infection with** *T. cruzi*[*] | 24/29 (82.76%) | 4/17 (23.53%) | 4/21 (19.05%) | 6/21 (28.57%) | 38/88 (43.18%) | 31/69 (44.93%) |
| **CD can be treated with anti-parasitic medication** | 10/25 (40%) | 5/17 (29.41%) | 4/21 (19.05%) | 7/20 (35%) | 26/83 (31.33%) | 24/69 (34.78%) |
| ***T. cruzi* is transmitted through triatomine feces** | 11/29 (37.93%) | 3/17 (17.65%) | 5/21 (23.81%) | 6/21 (28.57%) | 25/88 (28.41%) | 22/69 (31.88%) |
| **Correctly identified potential symptoms of acute CD** | 8/29 (27.59%) | 6/17 (35.29%) | 6/21 (28.57%) | 4/21 (19.05%) | 24/88 (27.27%) | 22/69 (31.88%) |
| **Correctly identified all potential modes of** *T. cruzi* **transmission** | 2/28 (7.14%) | 1/14 (7.14%) | 2/17 (11.76%) | 1/18 (5.56%) | 6/77 (7.79%) | 6/69 (8.70%) |

[*]Asterisks indicate a significant difference (p<0.05).

**Table 3. Mean and median knowledge scores by national park.**

| National Park | Mean knowledge score (%) | Median knowledge score (%) | Number of respondents |
|---|---|---|---|
| **Big Bend National Park (BIBE)** | 5.95 (59.50%) | 6 (60%) | 22 |
| **Saguaro National Park** | 4.71 (47.10%) | 4 (40%) | 17 |
| **Organ Pipe National Monument** | 4.59 (45.90%) | 5 (50%) | 17 |
| **Southeast Arizona Group** | 4.86 (48.60%) | 4 (40%) | 13 |
| **Receives triatomine and/or Chagas Disease awareness/education material*** | | | |
| **Yes (BIBE)** | 5.95 (59.50%) | 6 (60%) | 22 |
| **No (All other parks)** | 4.68 (46.80%) | 4 (40%) | 47 |

*Asterisks indicate a significant difference (p<0.05).

employees knew significantly more about dogs becoming sick from infection with *T. cruzi* compared to OPNM employees (z = 3.24; p<0.01), SEAZ employees (z = 2.91; p = 0.01), and SAGU employees (z = 3.60; p<0.01). Additionally, BIBE employees (z = 2.99; p<0.01) and SAGU employees (z = -2.82; p = 0.01) had significantly greater knowledge of the triatomine bug's nocturnal behavior than OPNM employees.

Among the respondents that answered each question from all the knowledge, attitudes, and practices sections (n = 69), aggregate knowledge scores ranged from 0 to 10 with the mean knowledge score being 5.09 (50.90%) and the median knowledge score being 5 (50%). The Cronbach's alpha demonstrated that the knowledge scale was internally consistent (0.71) and could be used as a reliable measure of aggregate CD knowledge. BIBE had marginally higher average and median knowledge scores than the other parks, but the difference was not statistically significant when evaluated against parks individually (Table 3). However, when aggregating parks by provision of triatomine and CD awareness and educational materials, BIBE had a significantly higher average knowledge score than all other parks when compared by this measure of educational material availability (t = -2.38; CI: -2.35, -0.20; p = 0.02). We found no differences in aggregate knowledge scores or individual knowledge questions by pet ownership status or housing status. Knowledge scores were similar for the Arizona parks regardless of respondent gender, ethnicity, age, education, and length of employment with the NPS. Knowledge scores were similar between the Arizona parks and BIBE regardless of whether employees lived in NPS housing, and the average amount of hours worked outdoors per week.

## NPS employee housing associated with increased triatomine exposure risk

The majority of NPS employees reported seeing triatomines during their employment with the NPS (n = 60; 67.42%). Among these employees, approximately half found the insects within their home (n = 29; 48.33%). The number of employees who correctly identified the triatomine bug in the knowledge section was 76 (86.36%). When restricting analysis to only these respondents, 68.42% (n = 52) reported seeing triatomines during their employment, of which 53.85% (n = 28) also reported finding triatomines within their homes. Within this subset of employees who reported seeing triatomines during their employment (n = 52), BIBE had the highest proportion of their employees reporting triatomines inside their homes (n = 14; 73.68%), while SAGU had the lowest proportion (n = 4; 36.36%). Only one (n = 1; 2%) employee that owned a pet reported finding triatomines in their pet's bedding, or in other areas where their pet spends time.

Approximately 1 out of 3 employees (36.36% (n = 32)) reported unexplained insect bites while sleeping. Among those bitten, most reported 1–5 bites a year (n = 23; 71.88%). The most common insect bite location was on the legs (n = 27; 84.38%) and arms (n = 21; 65.63%), while

21.88% (n = 7) reported bites on the face. Among the different parks, employees of SEAZ were the most likely to report unexplained insect bites while sleeping (n = 9; 52.94%), while SAGU employees reported the least (n = 3; 14.29%). In addition to triatomines, 75.28% of NPS employees reported small animals or rodents in or very near their home (n = 67). The most frequent locations of these small animals or rodents included the yard (n = 62; 92.54%), garage/carport (n = 26; 38.81%), and kitchen (n = 19; 28.36%).

NPS employees that lived in NPS housing had 10.32 higher odds of finding triatomines in their homes than NPS employees that did not live in NPS housing (95% CI: 2.41, 53.71; p = 0.04; p<0.01) and 2.97 higher odds of unexplained insect bites while sleeping (95% CI: 1.06, 8.80; p = 0.04). However, although NPS housing was not associated with small animal or rodent encounters (p>0.05), NPS employees who reported finding small animals or rodents in or very near their home had 6.26 higher odds of also reporting triatomines within their home (95% CI: 1.25, 43.10; p = 0.02).

## NPS employees feel at risk for CD and believe triatomine bug intervention is important

The negative perceptions of CD severity among NPS employees ranged from moderate to somewhat high, but anxiety regarding the disease did not discourage them from working with the NPS. For instance, while 35.96% (n = 32) of employees from the entire sample felt that CD was a serious illness in the surrounding area, only 14.61% (n = 13) agreed that CD negatively affected their feelings of working for the NPS. Additionally, 58.43% (n = 52) of employees felt at risk for developing CD while working for the NPS and two-thirds indicated that they would seek medical advice following a triatomine bite (66.29%; n = 59). Most employees felt that triatomine bug control was important (83.15%; n = 74). No SAGU employees felt CD negatively affected their work with the NPS. BIBE employees agreed significantly more than OPNM employees (z = -2.77; p = 0.02) and SEAZ employees (z = -2.67; p = 0.02) that triatomine bug control was important to them (Fig 2).

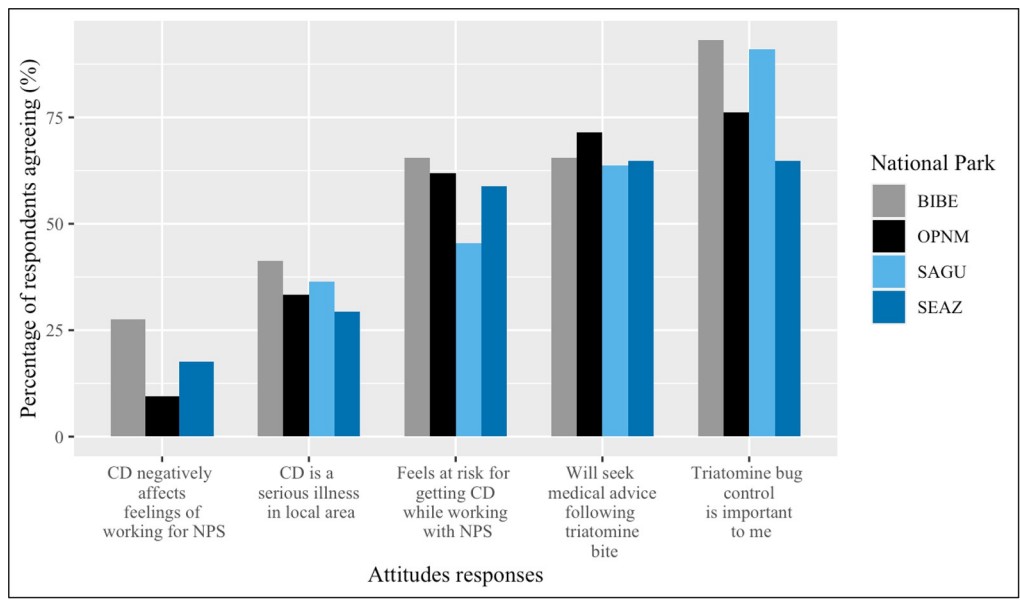

**Fig 2. Proportion of respondents agreeing with attitudes statements by national park.** Asterisk (*) indicates a significant difference between BIBE, OPNM, and SEAZ respondents (p<0.05).

We explored Principal Components Analysis (PCA) as a method to reduce dimensionality of the attitude items. The assumptions to conduct PCA were met as shown by a Kaiser-Meyer-Olkin value of 0.63 and a significant Bartlett's test of sphericity ($p < 0.0001$). All items in the attitudes section had factor loadings higher than 0.70 and communalities of 1.10 or lower, resulting in two extracted factors. The two extracted factors measured latent constructs of *perceived anxiety regarding CD* (CD negatively affects my feelings of working for the NPS, CD is a serious illness in the local area, and I am at risk for getting CD while working for the NPS) and *personal agency to address CD risk* (I will seek medical advice following a triatomine bug bite and triatomine bug control is important to me), shown in S2 Table. The Cronbach's alpha for perceived anxiety regarding CD (alpha = 0.64) reflected an internally consistent measure, while personal *agency to address CD risk* (alpha = 0.49) was marginal.

There was no significant difference in the factor values among the different parks. While *personal agency to address CD risk* was similar for the parks regardless of housing type, NPS employees who lived in NPS housing had significantly higher *perceived anxiety of CD* than those who lived in non-NPS-housing (t = -3.38; 95% CI: -1.08, -0.28; p = 0.001). *Perceived CD anxiety* was also positively associated with *personal agency to address CD risk* (n = 87; r = 0.27; p = 0.01), but it was not associated with knowledge of CD and triatomine vectors. *Personal agency to address CD risk* was not significantly associated with knowledge scores.

We found no differences in individual attitudes questions, *perceived anxiety of CD*, or *personal agency to address CD risk* based on pet ownership status. We found no differences in *personal agency to address CD risk* based on housing status.

## Employees from all parks have similar triatomine preventive practices

Among the respondents that answered all knowledge, attitudes, and practices questions (n = 69), aggregate practice scores ranged from 0 to 36 with the mean practice score being 25.25 and the median practice score being 25. The Cronbach's alpha (0.71) supported the reliability of the scale and our measure of aggregate practices. SAGU had a higher mean and median practice score than the other parks, but the differences were not statistically significant, as shown in Table 4. Additionally, respondents from parks that distributed triatomine and CD awareness and educational materials did not have significantly different practice scores than parks that did not distribute such materials. Practice scores were similar for the Arizona parks regardless of respondent gender, ethnicity, education, age, and length of employment with the NPS. Moreover, practice scores were similar between the Arizona parks and BIBE regardless of housing status, pet ownership status, and the average amount of hours worked outdoors per week.

Prior to testing for associations between knowledge, PCA, and practice scores, we removed 1 outlier respondent from the practices section due to answering "never" for each question.

**Table 4. Mean and median practice scores by national park.**

| National Park | Mean practice score (%) | Median practice score (%) | Number of respondents |
|---|---|---|---|
| **Big Bend National Park (BIBE)** | 25.64 | 25.50 | 22 |
| **Saguaro National Park** | 26.47 | 27 | 17 |
| **Organ Pipe National Monument** | 25.71 | 24 | 17 |
| **Southeast Arizona Group** | 22.38 | 24 | 13 |
| **Distributes triatomine and/or Chagas disease awareness/education material** | | | |
| **Yes (BIBE)** | 25.64 | 25.50 | 22 |
| **No** | 25.06 | 25 | 47 |

Knowledge scores were not correlated with practice scores. However, there was a weak, positive correlation between *personal agency to address CD risk* and practice scores (n = 86; r = 0.24; p = 0.02). Interestingly, there was a weak, negative association between *perceived anxiety of CD* and practice scores (n = 86; r = -0.21; p = 0.05).

The most common control/preventive measures used by NPS employees at the time of the survey included reducing outdoor light (n = 54; 61.36%) and creating a vegetation/debris free zone around their home (n = 29; 32.95%). Only 1 respondent indicated that they slept under a bed net (1.14%). Approximately one third of employees indicated they do nothing for triatomine bug prevention (n = 27; 30.68%). In contrast, the most common control/preventive measures that NPS employees would consider using, but do not currently employ, were natural pesticides (n = 65; 73.86%) and creating vegetation/debris free zones around their homes (n = 62; 70.45%), while the least considered measures were nothing (n = 5; 5.68%) and using bed nets (n = 21; 23.86%).

NPS employees generally implemented preventive practices at similar frequencies across the different parks. As indicated by the proportion of respondents selecting often or always, filling cracks or crevices in the home (n = 30; 43.48%) and removing debris from the yard (n = 31; 44.93%) were the least frequently practiced behaviors, while keeping wood piles at least 10 ft away from the house (n = 59; 85.51%) and turning off inside and outside lights at night (n = 56; 81.16%) were the most frequently practiced behaviors. Significantly more SAGU employees reported often or always having cracks or crevices in their home filled compared to SEAZ employees (p = 0.02, Fig 3); however, these results could have been influenced by the fact that more SAGU employees (95.45%) live in non-NPS housing than SEAZ employees (41.18%).

We did not find any significant associations between most preventive practices and finding kissing bugs inside the home. However, we found an association between ensuring window

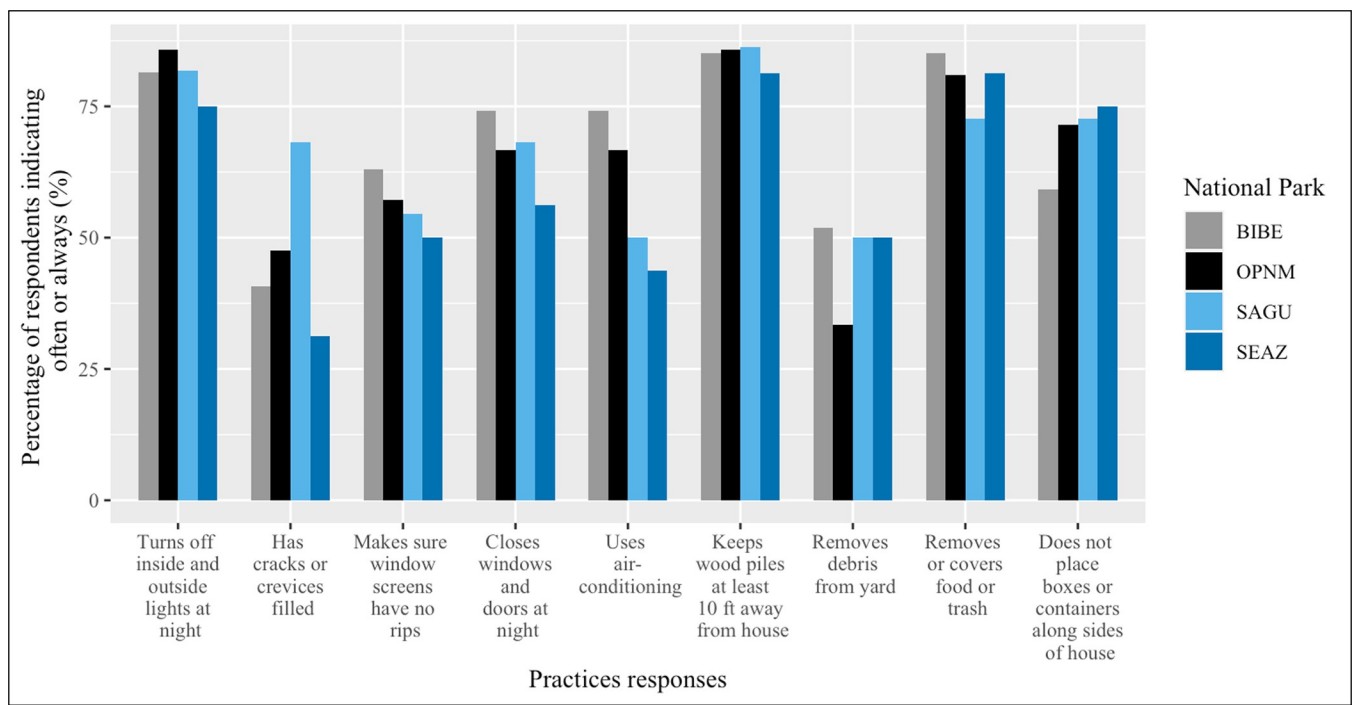

**Fig 3. Proportion of respondents indicating often or always implementing practices by national park.** Asterisks (*) indicates a significant difference between parks (p<0.05).

screens were repaired and reporting unexplained insect bites at night. Employees who indicated that they often or always made sure window screens had no tears had a 79% reduction in the odds of reporting unexplained insect bites at night than employees who indicated rarely or never for the same question (95% CI: 0.04, 0.93; p = 0.02).

Following exposure to a triatomine bug, the CDC recommends submitting the bug for testing and visiting your primary care provider (PCP) if you are worried about CD [40]. However, in this study, following a triatomine bite, only 35.96% (n = 32) would submit the bug for testing and two-thirds indicated that they would seek medical advice following a triatomine bite (66.29%; n = 59). The least common actions in response to a triatomine bite were doing nothing (n = 2; 2.25%), visiting the emergency room (n = 3; 3.37%), and visiting an infectious disease specialist (n = 6; 6.74%).

With regards to information on triatomines, most employees received this information from the park safety officer/NPS biologist (n = 51; 57.30%) and the CDC (n = 32; 35.96%), while the least common information sources were social media (n = 5; 5.62%) and the state health department (n = 11; 12.36%). Additionally, 17.98% (n = 16) reported not looking up CD information.

## Discussion

Among the southwestern NPS units, employees are frequently exposed to triatomines and small animals or rodents. More than half of the respondents in our survey reported seeing triatomines during their NPS employment, and more than half of those reports were within their homes. In this survey, over one-third of respondents reported unexplained insect bites at night. Similar to findings from a survey conducted in Arizona [20], we found that the arms and legs were the most frequently reported anatomical bite sites, rather than the face. SAGU had the smallest proportion of employees reporting unexplained insect bites at night and seeing triatomines in their home and also had the highest proportion of employees living in non-NPS housing. In contrast, BIBE had the highest proportion of employees reporting seeing triatomines in their home and the highest proportion of employees living in NPS housing. NPS housing is associated with exposure to triatomines, which demonstrates a risk for the potential transmission of *T. cruzi* to NPS employees who live in federal housing. This may highlight unique aspects of living in a park environment, potential flaws in NPS housing quality, barriers to maintenance and repair requests, or regional park differences. The lack of association between many prevention practices and exposure to triatomines may also suggest that other factors are important to CD risk mitigation. It may also reflect a lack of concern among employees, though this is unlikely since most employees reported that triatomine bug control is important to them. The recent funds dedicated to repairing and replacing aging infrastructure as part of the American Infrastructure Act are an important step towards reducing CD risk in NPS housing.

Despite the high frequency of human-triatomine bug exposure in southwestern national parks, NPS employees had low overall knowledge of CD. Knowledge gaps were related to *T. cruzi* transmission routes, acute CD signs and symptoms, and CD treatment. A lack of knowledge in these areas can lead to fewer individuals seeking medical attention or receiving treatment [41,42]. We found that respondents from NPS park units that made educational and awareness materials available had significantly higher knowledge scores than those from parks that did not distribute such materials. These results are similar to those reported by Granados and others [42] who found that US physicians who participated in a CD educational webinar reported greater knowledge of CD following the webinar. Interestingly, we did not find a difference in knowledge scores for pet owners versus non-pet owners, despite high exposure of

dogs to *T. cruzi* in this geographic region [43]. Perhaps, veterinarians can take a more active approach of educating dog owners in this area on CD risk.

Increased knowledge scores were not associated with increased preventive practices. Although focused on a different vector-borne disease, a KAP study conducted in Greenbelt Park also found that NPS employee knowledge of Lyme disease was not associated with tick preventive measures. Additionally, two studies conducted in Brazil and Ecuador reported that knowledge of triatomines was not associated with decreased triatomine bug infestation in homes [44–46]. Therefore, CD educational campaigns that solely rely on disseminating information may be ineffective at increasing triatomine bug preventive behavior among NPS employees [47].

In addition to knowledge barriers, the NPS should also consider NPS employees' perceptions of disease susceptibility and severity [48]. In general, employees felt susceptible to acquiring CD, but few felt that it was a severe or serious disease. This may represent risk normalization, whereby NPS employees minimize the threat of CD despite their high exposure to CD-vectors. Alternately, this may reflect the current understanding that autochthonous CD is a relatively rare risk. For example, research estimates that an average of 900 to 4,000 contacts with infected triatomines are required for human infection [49]. If NPS employees do not perceive CD as a serious threat, they may be unlikely to change their behavior. We also found personal agency to address CD risk was positively associated with practice scores, and perceived anxiety of CD was positively associated with personal agency to address CD risk. These results suggests that greater anxiety of CD may increase preventive behaviors and personal agency among NPS employees. Increases in perceived risk and fear were also found to increase the frequency of preventive behaviors regarding the SARS-CoV-2 virus [50].

However, using only fear-based approaches to increase perceived disease severity and susceptibility is not sufficient to increase the adoption of preventive behaviors [51,52]. The Extended Parallel Process Model (EPPM) is a framework to help identify how individuals will respond to fear-based messaging considering their perceived severity/susceptibility of a disease and self-efficacy. Based on the EPPM, fear-based approaches that do not also aim to improve self-efficacy can lead to maladaptive behaviors [52]. We found that employees who lived in NPS housing had significantly higher perceived anxiety of CD than those who lived in non-NPS housing, but not higher personal agency or practice scores. It may be that employees living in NPS housing have an increased interaction with triatomines that generated more fear, but this is not coupled with higher self-efficacy to complete interventions; additional data is needed to draw conclusions in this area.

More than half of all employees from each park reported engaging in all triatomine bug prevention practices often or always. Thus, NPS employees are frequently doing behaviors to prevent triatomines. However, the most frequently practiced behaviors were ones that required the least amount of physical exertion, such as turning off the lights at night, whereas the least frequently practiced behaviors required more effort, like having cracks or crevices filled. Therefore, NPS employees may have decreased self-efficacy for those behaviors that require more effort or for those which NPS tenants do not have control or authority to perform. However, turning off the lights at night may have been performed for natural resource outcomes or night sky policies rather than CD prevention.

In order to increase self-efficacy and preventive behaviors, communication should emphasize the effectiveness and simplicity of protective behaviors at preventing exposure to triatomines and potential CD transmission [51,53]. For instance, we found that employees who indicated they often or always made sure window screens lacked tears had a 79% reduction in the odds of reporting unexplained insect bites at night. These results can be used in messaging or to direct maintenance work flows to enable resources dedicated to screen repair. Fewer

insect bites at night may also bring peace of mind and comfort to NPS employees, both of which are universal motivators [54]. Targeting universal motivators in messaging makes the messages more compelling, which may make individuals more likely to engage in beneficial behavior [54].

The NPS can also communicate the feasibility of the behaviors that require more effort and provide specific instructions for more difficult behaviors. For example, clearly describing the steps for employees to fill out repair forms for filling cracks in walls during onboarding may be one such intervention. We also found that significantly more SAGU employees had cracks or crevices filled than SEAZ employees, and more SAGU employees lived in non-NPS housing than SEAZ employees. This may suggest living in NPS housing is a barrier to having cracks or crevices filled and can be further examined by the NPS.

Although we included a "Not sure" response in the survey, several respondents skipped questions in the knowledge section, which further limited the number of responses available for certain analyses. In addition, the factors generated by PCA did not have high internal consistency measures. It may be beneficial to include additional questions that directly measure specific attitudes to offer a more robust PCA in future surveys. Additionally, the questions that asked respondents whether they experienced unexplained insect bites is a proxy for assessing triatomine bug bites. It is possible that this variable may not actually capture triatomine bug bites nor insect bites, generally. Lastly, we asked one contact from each park whether they distributed CD educational materials, but our survey did not include a question where respondents could indicate whether they received CD education. This means that despite BIBE distributing educational material in the form of email advisories, respondents may not have read the emails. Therefore, we cannot assume that distributing educational materials contributed to greater knowledge scores.

In the US, the CD transmission cycle is not well understood, but human behavior and knowledge are deeply rooted within the cycle. Our results demonstrate the influence of knowledge and attitudes regarding CD on triatomine prevention behavior from a potential high-risk population in the US. Our work adds to the growing body of literature on the CD knowledge, attitudes, and practices of a potential high-risk population in the US.

## Supporting information

**S1 File. Knowledge, attitudes, and practices survey.**
(DOCX)

**S1 Table. KAP survey responses.**
(DOCX)

**S2 Table. Principle component analysis for attitudes items.**
(DOCX)

## Acknowledgments

We thank Thomas Athens from BIBE, Rijk Morawe from OPNM, JoAnne Blalack from SEAZ, Jeffery Conn from SAGU, and Joel Skarnikat from SAGU for their assistance with survey distribution. A special thanks to Stefanie Campbell, Dr. Maria Said, and members of the Harrington Lab at Cornell University for their assistance with the initial survey design and beta testing. The authors take full responsibility for the content of this work. The conclusions herein do not necessarily represent the official views of the Centers for Disease Control and Prevention or the Department of Health and Human Services.

## Author Contributions

**Conceptualization:** Antonio Alvarado, Emily M. Mader, Danielle Buttke, Laura C. Harrington.

**Formal analysis:** Antonio Alvarado.

**Funding acquisition:** Laura C. Harrington.

**Investigation:** Antonio Alvarado.

**Methodology:** Antonio Alvarado, Emily M. Mader, Laura C. Harrington.

**Project administration:** Emily M. Mader.

**Resources:** Laura C. Harrington.

**Supervision:** Danielle Buttke.

**Writing – original draft:** Antonio Alvarado.

**Writing – review & editing:** Emily M. Mader, Danielle Buttke, Laura C. Harrington.

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
