## [Decision Letter · Decision Letter 0]

23 Jun 2022

Dear Mr. Alvarado,

Thank you very much for submitting your manuscript "Southwestern National Park Service Employee Risk, Knowledge, and Concern for Triatomine Exposure: A Qualitative Analysis Using a Novel Knowledge, Attitudes, and Practices Survey" for consideration at PLOS Neglected Tropical Diseases. As with all papers reviewed by the journal, your manuscript was reviewed by members of the editorial board and by several independent reviewers. In light of the vastly differing reviews (below this email), we would like to invite the resubmission of a significantly-revised version that takes into account the reviewers' comments. 

We cannot make any decision about publication until we have seen the revised manuscript and your response to the reviewers' comments. Your revised manuscript is also likely to be sent to reviewers for further evaluation.

Sincerely,

Rhoel Ramos Dinglasan

Associate Editor

Walderez Dutra

Deputy Editor

Reviewer's Responses to Questions

**Key Review Criteria Required for Acceptance?**

**Methods**

-Are the objectives of the study clearly articulated with a clear testable hypothesis stated?

-Is the study design appropriate to address the stated objectives?

-Is the population clearly described and appropriate for the hypothesis being tested?

-Is the sample size sufficient to ensure adequate power to address the hypothesis being tested?

-Were correct statistical analysis used to support conclusions?

-Are there concerns about ethical or regulatory requirements being met?

Reviewer #1: Yes

Reviewer #2: Are the objectives of the study clearly articulated with a clear testable hypothesis stated? 

No, the methodology used has many deficiencies and there was no control of biases derived from the methodology.

-Is the study design appropriate to address the stated objectives

the study design only gives it a descriptive scope, the lack of randomization and a possible imposition on the participants to increase the number of participants in the study, can invalidate the results. Additionally, the methodology is not adequate to compare between groups.

-Is the sample size sufficient to ensure adequate power to address the hypothesis being tested?

the number of participants is adequate, however, it is not clear whether participation is entirely voluntary

-Were correct statistical analysis used to support conclusions?

The authors perform routine statistical analyzes and those that are done in similar studies, however, it is better to adopt other methodologies such as semi-structured interviews, which allow qualitative analysis. Additionally, not using triatomine samplers does not allow the participant to differentiate between reduced hematophagous, predatory and phytophagous... This possible memory bias is key not controlled, it is key to have wrong results

-Are there concerns about ethical or regulatory requirements being met?

there is no such concern,is ok

**Results**

-Does the analysis presented match the analysis plan?

-Are the results clearly and completely presented?

-Are the figures (Tables, Images) of sufficient quality for clarity?

Reviewer #1: Yes, the results were appropriate and rigourously collected.

Reviewer #2: -Does the analysis presented match the analysis plan?

yes, they represent the analysis plan

-Are the results clearly and completely presented?

yes, they are clearly present

-Are the figures (Tables, Images) of sufficient quality for clarity?

Table 1 it is innecesary 

at line 272, the authors said " NPS employees had low knowledge of CD and triatomine vectors". However, in the table 2 we found a differnt datas.. Correctly identified triatomine: 96,55%,82.35%, 76, 19%...

The question "The presence of animal reservoirs, such as pack rats, are important factors for human risk of CD". The authors have no evidence if this species plays a role as reservoirs in this ecosystem

**Conclusions**

-Are the conclusions supported by the data presented?

-Are the limitations of analysis clearly described?

-Do the authors discuss how these data can be helpful to advance our understanding of the topic under study?

-Is public health relevance addressed?

Reviewer #1: Yes, conclusions were appropriate based on the findings.

Reviewer #2: -Are the conclusions supported by the data presented?

no, the authors cannot be clear whether the outcome they evaluated (recognition of the vector) was not influenced by a reporting or recall bias. It is clear that the people evaluated live in areas with potential transmission, but there are other methodologies that allow to see if there is contact between people and triatomines and potential reservoirs in a particular environment.

-Are the limitations of analysis clearly described?

biases and controls were not reported

-Do the authors discuss how these data can be helpful to advance our understanding of the topic under study?

no, 203 / 5.000

Resultados de traducción

many of the conclusions of this study have been reported previously. currently, there are new methodologies and other types of approaches to know the risk that employees of NPS

-Is public health relevance addressed?

Studies that visualize the potential risk of people living in areas with the presence of enzootic transmission in the United States are interesting, however, more robust approximations must be made. Unfortunately, the methodology used does not allow knowing this

**Editorial and Data Presentation Modifications?**

Reviewer #1: Accept

Reviewer #2: (No Response)

**Summary and General Comments**

Reviewer #1: Well written article that contributes to the scientific literature.

Reviewer #2: These types of studies are important and necessary, however, the methodology adopted and the lack of bias control do not allow knowing the potential risk of infection. The results show that there is an apparent ignorance of the employees of the potential risk of vector and oral transmission. However, more appropriate outcomes (triatomine collection, photo taking, use of apps) are recommended. It is better to use semi-structured interviews with key informants and to carry out qualitative analyses.

PLOS authors have the option to publish the peer review history of their article (what does this mean?). If published, this will include your full peer review and any attached files.

Reviewer #1: No

Reviewer #2: No
---

## [Decision Letter · Decision Letter 1]

16 Aug 2022

Dear Mr. Alvarado,

We are pleased to inform you that your manuscript 'Southwestern National Park Service Employee Risk, Knowledge, and Concern for Triatomine Exposure: A Qualitative Analysis Using a Novel Knowledge, Attitudes, and Practices Survey' has been provisionally accepted for publication in PLOS Neglected Tropical Diseases.

Best regards,

Rhoel Ramos Dinglasan

Academic Editor

Walderez Dutra

Section Editor

Reviewer's Responses to Questions

**Key Review Criteria Required for Acceptance?**

**Methods**

-Are the objectives of the study clearly articulated with a clear testable hypothesis stated?

-Is the study design appropriate to address the stated objectives?

-Is the population clearly described and appropriate for the hypothesis being tested?

-Is the sample size sufficient to ensure adequate power to address the hypothesis being tested?

-Were correct statistical analysis used to support conclusions?

-Are there concerns about ethical or regulatory requirements being met?

Reviewer #1: (No Response)

**Results**

-Does the analysis presented match the analysis plan?

-Are the results clearly and completely presented?

-Are the figures (Tables, Images) of sufficient quality for clarity?

Reviewer #1: (No Response)

**Conclusions**

-Are the conclusions supported by the data presented?

-Are the limitations of analysis clearly described?

-Do the authors discuss how these data can be helpful to advance our understanding of the topic under study?

-Is public health relevance addressed?

Reviewer #1: (No Response)

**Editorial and Data Presentation Modifications?**

Reviewer #1: (No Response)

**Summary and General Comments**

Reviewer #1: (No Response)

PLOS authors have the option to publish the peer review history of their article (what does this mean?). If published, this will include your full peer review and any attached files.

Reviewer #1: No

---

## [Editor Report · Acceptance letter]

26 Aug 2022

Dear Mr. Alvarado,

We are delighted to inform you that your manuscript, "Southwestern National Park Service Employee Risk, Knowledge, and Concern for Triatomine Exposure: A Qualitative Analysis Using a Novel Knowledge, Attitudes, and Practices Survey," has been formally accepted for publication in PLOS Neglected Tropical Diseases.

Best regards,

Shaden Kamhawi

co-Editor-in-Chief

Paul Brindley

co-Editor-in-Chief
